# Tumour Derived Extracellular Vesicles: Challenging Target to Blunt Tumour Immune Evasion

**DOI:** 10.3390/cancers14164020

**Published:** 2022-08-20

**Authors:** Tatiana Lopatina, Alessandro Sarcinella, Maria Felice Brizzi

**Affiliations:** Department of Medical Sciences, Turin University, 10126 Turin, Italy

**Keywords:** extracellular vesicles, exosomes, tumour immune editing, tumour immune suppression, cell-to-cell communication, tumour antigens

## Abstract

**Simple Summary:**

Tumour onset and development occur because of specific immune support. The immune system, which is originally able to perceive and eliminate incipient cancer cells, becomes suppressed and hijacked by cancer. For these purposes, tumour cells use extracellular vesicles (TEVs). Specific molecular composition allows TEVs to reprogram immune cells towards tumour tolerance. Circulating TEVs move from their site of origin to other organs, preparing “a fertile soil” for metastasis formation. This implies that TEV molecular content can provide a valuable tool for cancer biomarker discovery and potential targets to reshape the immune system into tumour recognition and eradication.

**Abstract:**

Control of the immune response is crucial for tumour onset and progression. Tumour cells handle the immune reaction by means of secreted factors and extracellular vesicles (EV). Tumour-derived extracellular vesicles (TEV) play key roles in immune reprogramming by delivering their cargo to different immune cells. Tumour-surrounding tissues also contribute to tumour immune editing and evasion, tumour progression, and drug resistance via locally released TEV. Moreover, the increase in circulating TEV has suggested their underpinning role in tumour dissemination. This review brings together data referring to TEV-driven immune regulation and antitumour immune suppression. Attention was also dedicated to TEV-mediated drug resistance.

## 1. Introduction

Tumour-derived extracellular vesicles (TEV), are small membrane vesicles carrying a specific set of proteins, nucleic acids, and lipids designed to spreading tumour signals and establishing a favouring tumour microenvironment (TME). It was widely reported that TEV from different cancers regulate the immune system by acting on different cell types. Moreover, TEV express markers of the original tumour cell, thereby representing non-invasive circulating cancer biomarkers [1,2]. Since the immune reaction plays a permissive role in tumour growth and metastasis, the switch of the immune program from anticancer to tumour tolerance has generated great interest. 

The anticancer immune defence is mainly based on tumour-specific cytotoxic T lymphocytes (CTL) [3,4,5]. CTL differentiation occurs in secondary lymphoid organs once naive T-cells undergo tumour-specific stimulation from antigen-presenting cells (APC). APC are mainly represented by dendritic cells (DC), but also macrophages, B-cells and others [6,7]. The maturation/activation state of APC dictates naive T-cell fate: effector CTL or exhausted T-cells granting tolerance. It was shown that TEV are primarily internalised by APC, thereby suggesting that TEV regulate the immune response via APC reprogramming. In this review, data on tumour immune editing mediated by TEV released from cancer cells and tumour-associated cells are discussed.

## 2. TEV and CTL Reprogramming

In healthy subjects, differentiated tumour-specific effector CTL enter the circulation, infiltrate the tumour, and attack tumour cells by interacting with a specific tumour antigen. CTL cytotoxicity is mediated by the release of pro-inflammatory cytokines, perforin and granzymes, or by direct contact with apoptosis-inducing molecules expressed by tumour cells [8,9]. Tumour antigen presentation by APC is crucial for CTL differentiation, while insufficient to sustain the cytotoxic immune response [10]. To drive specific antitumour activity, T-cells must be activated by “mature” APC. During differentiation, APC “capture” invading pathogens [11,12], neo-antigens [13], apoptotic cells [14] and EV [15]. These signals dictate APC specification towards pro-inflammatory or tolerant phenotypes, resulting in a specific T-cell commitment [16]. Since the role of DC in orchestrating the immune response has been extensively investigated, the mechanisms driving tumour tolerance described herein mainly refer to DC-mediated T-cell differentiation. The high expression of MHC, CCR7, co-stimulatory factors CD80, CD83 and CD86 identifies pro-inflammatory DC, also named “mature” DC [17,18,19]. Conversely, the low expression of these markers defines “immature” DC, which maintain the immune tolerance, suppress immune cells-mediated attack, and prevent autoimmunity. Physiologically, “immature” DC differentiation occurs during apoptosis, since apoptotic cells inhibit the expression of co-stimulatory molecules [14,20,21,22,23,24]. 

TEV hold endogenous tumour antigens; therefore, they can be considered for potential use in an anticancer vaccine [25]. However, they mainly act as immune suppressors. Several studies have demonstrated that in vivo TEV are mostly internalised by myeloid cells [26,27,28]. Once stimulated with TEV, DC, displaying an immature phenotype, prevent the immune attack and stimulate the expression of immune checkpoint molecules [7,27,29,30]. For example, Cytotoxic T Lymphocyte Antigen-4 (CTLA4) and Programmed cell Death protein 1 (PD-1) induce CTL anergy or promote regulatory T-cell differentiation (Treg) [31,32,33,34,35,36]. DC interact with thousands of T cells per hour [37], while their MHC^+^ EV can cross-activate additional T cells and amplify the “onco-message” [38,39,40]. Moreover, exhausted T cells [41], Treg [42,43] and other tumour-infiltrating immune cells release their own EV, further suppressing the immune response. Apart from tumour-infiltrating DC [44], the role of other myeloid cells (monocytes and macrophages) in tumours is extensively documented [45,46,47]. Indeed, macrophages trapping TEV can transfer their oncogenic cargo to fibroblasts and endothelial cells, resulting in cancerisation of adjacent tissue [48].

Since tumour-specific T-cell differentiation depends on the maturation state of APC, and APC are primary TEV targets, it is conceivable that TEV interaction with peripheral and tissue resident APC may be considered a driving event in tumour immune editing. 

## 3. TEV Characterisation 

Tumour cells release different subpopulations of TEV according to their biogenesis and origin: microvesicles, exosomes, apoptotic bodies and oncosomes, to name a few. Microvesicles bud directly from the plasma membrane and are 0.1–1 μm in size. Exosomes include EV with different functions, usually formed from multivesicular bodies and with a diameter ranging from 30–120 nm. Dying cells release vesicular apoptotic bodies (50 nm–2 μm) that play a role in autoimmunity [49,50,51]. In cancer, membrane shedding can also give rise to large EV, termed oncosomes (1–10 μm) [52,53]. Oncosomes transfer tumour-promoting factors as oncoproteins, oncomiRNAs and tumour DNA [54]. In this review, the term EV will refer to a mixed population of small vesicles unless otherwise stated by the studies. The EV subpopulation shares several markers, including Toll-like receptors (TLRs), pathogen-associated molecular patterns (PAMPs) and internal damage-associated molecular patterns (DAMPs) [55,56,57,58], involved in myeloid cell targeting and immune regulation. Herein, we describe TEV marker participation in tumour immune suppression (Figure 1 and Table 1).

### 3.1. Tetraspanins 

Tetraspanins (CD63, CD81, CD9, etc.) organise membrane tetraspanin-enriched microdomains involving selected transmembrane and cytosolic signalling proteins [59]. The tetraspanin-mediated sorting of specific adhesion receptors may influence EV uptake. For example, the adhesion molecules ICAM-1/LFA1 are involved in EV uptake by leukocytes [60,61,62]. Moreover, tetraspanins interact with ADAM10 and ADAM17 and control their sheddase activity [63]. In the endosomal compartment, active ADAM10 and ADAM17 enriched in EV cleave preproteins to release soluble molecules, as shown for TNFα in ovarian cancer and melanoma [64]. Tetraspanins also regulate sorting of MHC I and MHC II in EV [55], thereby regulating TEV immunogenicity. 

Several studies also provide evidence for the role of CD81 in the assembly of functional immunological synapses acting as T-cell co-stimulatory signals [65]. Therefore, tetraspanins, besides activating naive T-cells via the MHC complexes, also act as co-stimulatory signals. 

### 3.2. Toll-Like Receptors (TLRs) 

TLRs belong to a family of pattern recognition receptors, which play a critical role in the innate immune responses by recognising several ligands also carried by TEV [66,67]. In cancer, these ligands include PAMPs, double-stranded RNA and/or DNA [68], and internal DAMPs, as fibrinogen, nuclear and cytosolic proteins, and Heat Shock Proteins (HSP) [69,70,71,72]. TLR1-2 and TLR4-6 proteins are localised on the surface membrane, whereas TLR3 and TLR7–9 are accumulated in the endosome or lysosome compartment, as well as in the endoplasmic reticulum [73]. Immune cells, epithelial [74], endothelial [75] and tumour cells [76] express TLRs [77]. TLR stimulation by TEV activates both the innate and adaptive immune responses and induces inflammatory signals in target cells. In cancer, TLR activation triggers an intracellular cascade, resulting in NF-κβ activation and the expression of genes involved in immune suppression [8,78]. Moreover, the expression of TLRs on TEV suggests that they can bind and transfer specific ligands to target cells, further impairing the antitumour immune response [39]. 

### 3.3. Heat Shock Proteins (HSP)

HSP are involved in antiapoptotic signalling [79,80], EMT [81], tumour angiogenesis and invasion [82], as well as in drug resistance [83,84]. Immune cells express HSP receptors as TLRs, leading to TEV endocytosis, antigen cross-presentation, T-cell cross priming, and activation of the immune response. HSP90, HSP70, HSP27 are DAMPs members expressed by EV derived from cancerous [85] and non-cancerous cells. However, the number of circulating HSP70^+^ EV is higher in cancer patients than in healthy donors [85] and is correlated with metastases growth [86]. Proteomics revealed that TEV are enriched in HSP90 [87], HSP72 [88], HSP105 [89], HSP70 [90,91] and HSP86 [92]. These observations have suggested that for HSP soluble forms, TEV can act as immunoregulators. It has been shown that the interaction of HPS expressed by TEV and TLR2 or TLR4 on myeloid cells (DC or MDSC) triggers pSTAT3 expression and IL-6 production, thereby repressing the antitumour immune reaction [88,90]. Moreover, TEV released upon hypoxia were found to be enriched in HSP70 and HSP90 and involved in invasiveness and stemness properties of prostate cancer cells [91]. In addition, human-melanoma-derived TEV enriched in HSP86 interact with TLR4 on myeloid cells, resulting in PDL-1 upregulation and blockade of T-cell activation [92]. 

Meanwhile, the antitumour activity of TEV enriched in different HSP isoforms was also reported [93,94,95,96,97]. HSP70^+^ TEV secreted by human pancreatic and colon carcinoma cell lines induced NK [93,94] and Th1 reactivity [95], and converted Treg into Th17 cells [96]. This resulted in a strong antitumour immune response. Pretreatment of melanoma and colon carcinoma cells with recombinant HSP70 increased the release of HPS70^+^ TEV, which, in vivo, translates in a strong anticancer immune response [97].

The correlation of different circulating HSP in cancer patients with tumour progression [86], the relevance of HSP in immune regulation [98], as well as their enrichment in TEV [84] strongly support the contribute of HSP^+^ TEV in mediating tumour immune suppression. However, further studies are required to clarify this specific topic. 

### 3.4. Integrins (ITGA/ITGB)

Integrins are critical receptors for extracellular matrix proteins supporting cell adhesion and driving cell migration. α1β1 and α2β1 integrins are major collagen receptors, whereas α5β1 and αvβ3 integrins preferentially bind fibronectin. Fibronectin interacting with TEV-associated integrins drives the invasive phenotype to recipient fibroblasts [99]. TEV enriched in integrins support tumour spreading and metastasis formation [100,101,102,103]. Extracellular matrix proteins can link TEV and target cells as a bridge to induce specific biological responses. Indeed, it was shown that the expression of α6β1, α6β4 and αvβ5 in TEV dictates cell target specificity and organotropic metastasis [100]. Moreover, it has been shown that EV integrin cargo is involved in immune cell recruitment [104,105], suggesting that integrins in TEV may be instrumental to immune cell homing and specific targeting. Overall, TEV integrin content regulates their uptake [106], identifies specific secondary tissues [100], and attracts immune cells [107,108]. 

CD47 (Cluster of Differentiation 47) is an integrin-associated protein that is overexpressed by tumour cells and TEV and plays a role in immune regulation [109,110]. CD47 binds the SIgnal-Regulatory Protein alpha (SIRPα) [111], able to deliver inhibitory signals to macrophages. It was shown that downregulation of CD47 in TEV enhanced macrophages-mediated cancer cell phagocytosis [112].

### 3.5. TEV as Immunosuppressive Shuttles 

TEV hold immune checkpoint molecules and anti-inflammatory cytokines. Physiologically, the checkpoint machinery controls T-cell activation, counteracting unnecessary tissue damage and autoimmunity. In cancer, TEV enriched in immune checkpoint ligands bind T-cell-associated cognate receptors, thereby reducing tumour killing. Several immune checkpoint pathways are referred to as crucial for anticancer therapeutic approaches, and among them PD1/PD-L1, CTLA4/B7, TIM3, Trail and Fas/FasL have received particular attention [113] (Figure 2).

### 3.6. Programmed Death Receptor 1 (PD-1) and Its Ligand (PD-L1)

PD-1/PD-L1 signalling controls and impairs inflammation. Activated T cells express PD-1 and secrete inflammatory factors, such as IFNγ and TNFα, that in turn, upregulate PD-L1 in surrounding cells as well as in EV [114,115,116,117]. Several tumours hijack the PD-1/PD-L1 pathway by inducing the expression of PD-L1 in the TME compartment. Persistent PD-1/PD-L1 binding leads to T-cell exhaustion, inhibition of proliferation and T-cell apoptosis. Conversely, Treg (CD4+ Foxp3+) bearing PD-1 promotes naive CD4+ T cell differentiation towards Treg [118]. Thus, PD-1 expression not only suppresses effector T-cell function, but also dictates T-cell shift towards an immunosuppressive phenotype.

The immunosuppressive role of PD-L1+ TEV was reported in head and neck squamous cell carcinoma [119], prostate cancer, melanoma [115] and non-small cell lung cancer [120]. PD-1 blocking antibodies have been shown to exert remarkable anticancer immune activity in PD-L1+ tumours [121]. However, TEV enriched in PD-L1 were described as both mediators of tumour immune evasion and predictors of anti-PD-1 treatment effectiveness [118,122]. 

### 3.7. CTLA-4/B7/CD152

As PD-1, CTLA-4 is broadly engaged during tumour immune evasion to suppress CD4+ effector T-cells and boost Treg activity [123]. After CD80/CD86 binding, CTLA-4 suppresses CD8+ T cell proliferation, cell cycle progression and cytokine (IL-2, IFN-γ) production [124]. CTLA-4 competes with CD28 for CD80/CD86 ligands. CD28 activates and induces a T-cell pro-inflammatory phenotype, while CTLA-4, displaying high affinity for CD80/CD86, guarantees T-cell suppression [125]. All at once, stimulation of CTLA-4 on Treg leads to FoxP3 upregulation, thereby improving Treg functions [126]. In fact, anti-CTLA-4 antibodies reduce the number of Treg at the tumour site, inhibit their negative regulatory effects, and enhance the antitumour immune response [127]. Therefore, CTLA-4 may exert pleiotropic effects on target cells depending on the presence of discrete markers of activation.

TEV carry both CTLA-4 and its ligands CD80/CD86 [127,128,129,130]. It has been proposed that the expression of CD80/CD86 in TEV interferes with CTL activity by binding to CTLA-4 [128,129]. Circulating TEV carrying CTLA-4 were also detected in cancer patients, and their concentration correlated with a poor outcome [127,130]. It has been proposed that CTLA-4 in TEV acts as a scavenger for the CD80/CD86 pro-apoptotic ligands, thereby protecting tumour cells. According to this possibility, TEV enriched in CTLA-4 regulate the PTEN/CD44 signalling pathway to promote proliferation, self-renewal and liver metastasis [131].

### 3.8. T-Cell Immunoglobulin Domain and Mucin Domain 3 (TIM-3)

TIM-3, a transmembrane protein expressed by activated immune cells, suppresses pro-inflammatory cytokine secretion [132] and CTL function [133], thereby mediating immune suppression. 

It was shown that TEV are enriched in TIM-3 [134] and that the level of TIM-3 on circulating TEV positively correlated with cancer progression [135]. Moreover, TEV can transfer TIM-3 to macrophages, driving their M2 differentiation [134].

### 3.9. Cluster of Differentiation 73 (CD73)

CD73 is a 5′-nucleotidase that converts AMP into the anti-inflammatory adenosine [136]. In cancer, free adenosine generated by CD73 inhibits cellular immune responses, thereby promoting tumour immune escape [137]. MDSC, Treg or DC expressing CD73 generate adenosine in tumours, which triggers the anti-inflammatory response via T-cells, macrophages, and natural killer cells (NK) [138,139]. Several studies have reported that the expression of CD73 in TEV also contributes to immunosuppression in different cancers [140,141]. TEV transfer CD73 to T-cells and inhibit their cell cycle entry and clonal proliferation by increasing adenosine [140]. Consistently, inhibition of TEV production or CD73 expression significantly inhibit tumour growth by restoring T-cell clonal proliferation [140]. It was also shown that CD73+ TEV upregulate the expression of IL-6, IL-10, TNFα and TGFβ1 in TAM by activating the NF-κB pathway [141]. Moreover, circulating CD73+ TEV increased the risk of lymph node metastases and was associated with a poor prognosis. In addition, loss of CD73+ TEV enhanced the effectiveness of anti-PD-1 treatment by rescuing CTL function [141]. 

### 3.10. Fas/FasL

Like PD-L1, CTLA-4 and CD73, Fas/FasL is hijacked by cancer cells to suppress the immune response and induce tumour immune evasion. FasL (CD95) is the most studied mediator of T-cell apoptosis [142]. Fas+ T cells undergo apoptosis upon binding to FasL, which is expressed in APC, tumours, and other cell types. It was also shown that IL-10, VEGF and prostaglandin E2 (PGE2) enhance the expression of FasL in tumour cells [143]. The expression of the Fas competitor, c-FLIP, in Treg and tumour cells determines their resistance to FasL-mediated apoptosis [144,145,146]. In addition, in cancer cells, the binding of c-FLIP to FasL+ EV interferes with the activation of caspase-8, thereby promoting tumour invasiveness and metastasis formation [147]. 

Circulating TEV carrying both Fas and FasL were detected in cancer patients [148]. It has been proposed that TEV expressing Fas may act as FasL scavenger, protecting tumour cells from death-receptor-mediated apoptosis [148], while FasL+ TEV act as inducers of T-cell apoptosis [149,150]. In addition, patient-derived FasL+ TEV were shown to induce caspase-3 cleavage, cytochrome c release and the loss of mitochondrial membrane potential in target T-lymphocytes [151]. 

### 3.11. TRAIL 

TRAIL (Apo2L) is a transmembrane protein homologous to FasL (Apo1) and belongs to the tumour necrosis factor (TNF) superfamily. Its extracellular domain can be proteolytically cleaved from the cell surface, thereby acting as a cytokine. TRAIL was shown to induce caspase-8-dependent apoptosis in tumour cells by binding to death receptors 4 or 5 (DR4, TRAIL-R1, or DR5, TRAIL-RII) [152,153,154]. DR4 and DR5 are generally highly expressed on tumour cells, and their downregulation correlates with resistance to TRAIL-induced apoptosis. However, TRAIL-mediated resistance depends on different mechanisms. For example, TRAIL binds the decoy receptors DcR1 [155] and DcR2. DcR2 has a truncated cytoplasmic domain which activates NF-κB [156], resulting in inflammatory and pro-survival gene transcription [156,157]. In TRAIL-resistant tumours, TRAIL activation promotes tumour cell migration and inflammatory immune responses partially via c-FLIP. Overexpression of cFLIP and mutation of caspase-8 rewire TRAIL signalling to promote NF-κB activation [158,159].

TRAIL carried EV exerting antitumour activity [160,161,162,163,164]. However, circulating TRAIL+ TEV were detected in head and neck cancer [164] and glioblastoma patients [130]. It can be hypothesised that cancer cells eliminate TRAIL by releasing TRAIL+ TEV, which in turn act as scavengers for death receptors. However, TRAIL+ TEV were shown to induce T-cell apoptosis [150]. In addition, TEV carrying the TRAIL receptor DR5 were shown to impair tumour apoptosis [165].

### 3.12. Protein Corona and TEV Mechanism of Action

Apart from membrane-associated proteins, such as PD-L1, CD73 and FasL, TEV are enriched in growth factors and cytokines involved in the immune regulation. These proteins can be part of “protein corona”, artificially linked to TEV [166,167,168], or naturally linked to their receptors, as shown for TGFβ1 [169]. 

Several membrane receptors expressed in TEV can bind their ligands (integrins/extracellular matrix proteins; TLRs/HSP and nucleic acids; receptors/growth factors and cytokines). Interestingly, EV covered with protein corona were more effective in monocyte activation, compared to “pure” EV or protein aggregates (protein corona). This suggests that EV-mediated immune cell activation depends on both protein corona and EV cargo. Conversely, it has been shown that DC differentiation and maturation only depend on EV cargo [167,168]. These findings indicate that TEV-mediated immune regulation is much more complex than expected.

### 3.13. TGFβ1 

TGFβ1 has been extensively investigated as an immunosuppressive factor in tumour settings. Webber and colleagues [169] showed the natural surface binding of TGFβ1 on TEV. TGFβ1, bound as a latent form, is able to activate the SMAD signalling pathway and gene expression in target cells [169]. TEV enriched in TGFβ1 were described in gastric [170], bladder [171], head and neck cancer [39], mesothelioma [172], breast [173], prostate and other cancers [169]. It was shown that TGFβ1+ TEV boost the expression of CTLA-4, FasL, IL-10 and other immunosuppressive proteins [174]. TGFβ1 acts on CTL by interfering with the expression of perforin, granzyme A and B, FasL and IFNγ [175], and by inducing Treg differentiation and IL-10 production [176,177]. 

### 3.14. Interleukin 10 (IL-10)

IL-10 is a regulatory cytokine with pleiotropic effects in various cell types [178]. IL-10 can directly inhibit T-cell responses [179,180], thereby enhancing Treg differentiation [177,181,182]. More importantly, IL-10 inhibits DC and macrophage functions by suppressing inflammatory cytokine production and inhibiting the expression of MHC class II and the co-stimulatory-molecules CD80/CD86 [183]. In cancer, the co-expression of IL-10 and tumour antigens reprogram APC towards a tumour-tolerant phenotype. 

As TGFβ1, TEV enriched in IL-10 were detected in plasma of different cancer patients; however, the most relevant action performed by TEV is the induction of IL-10 and TFGβ1 in target cells via their RNA content [184,185,186].

**Table 1 cancers-14-04020-t001:** Protein content of TEV involved in immune regulation and cellular effectors of each TEV marker. TEV mechanism of action in immune regulation is also indicated.

Molecules Enriched in TEV	Cellular Effectors	Immune-Related Mechanism	Refs.
Tetraspanins (CD9, CD63, CD81)	LFA-1	EV uptake by leukocytes	[60,61,62]
	MHC I, MHC II	Regulation of TEV immunogenicity	[55,65]
	ADAM10, ADAM17	ADAM sheddase activity	[63]
HSPs	TLRs	NF-κβ activation and expression of genes involved in immune suppression in tumour cells	[8,78]
		MDSCs-mediated production of immunosuppressive factor and induction of TAM/M2 polarisation	[85,88]
Integrins (ITGA/ITGB), CD47	Extracellular matrix proteins (fibronectin, collagen)	Boost of fibroblast invasive phenotype, tumour spreading and immune cell recruitment	[99,100,101,102,103,104,105,106,107,108,109,110,111,112]
PD-L1	PD-1	Suppression of CTL functions and induction CD4+ Treg activity	[118,122]
CD80/CD86	CTLA4		[123,124]
TIM-3	Galectin-9	Induction of TAM/M2 differentiation	[134,135]
CD73	AMP	Suppression of T-cell clonal expansion and production of immunosuppressive cytokines by M2/TAM	[140,141]
FasL	Fas	Induction of T-cell apoptosis	[149,150]
TRAIL	TRAIL-RI/RII		[150]
TGFβ	TGFβ-R1/2	Increased expression of immunosuppressive factors, suppression of CTL functions and induction of CD4+ Treg activity	[174,175,176]
IL-10	IL-10R	Suppression of CTL function, induction of CD4+ Treg activity and APC reprogramming towards tolerance	[178,179,180,181,182,183]

### 3.15. TEV Non-Coding RNA Content

EV in general, and TEV in particular, contain and transfer different types of RNAs (mRNA, microRNA, long non-coding RNA, circular RNA, rRNAs, tRNAs, PIWI-interacting RNAs, mitochondrial RNAs, Y RNAs and vault RNAs) to target cells [187,188,189] (Figure 3). TEV-RNA content was firstly described by Skog et al. [190]. Data from different studies [191,192,193] have suggested a crucial role of EV-RNA in the horizontal gene transfer and epigenetic regulation mediated by EV. Stable extracellular RNAs, lipids and proteins linked to EV have been found in all biological fluids [194,195], suggesting their role in several diseases, and in particular, in cancer [196].

It has been shown that TEV are enriched in microRNA with respect to their cell of origin. TEV miRNA content regulates and suppresses immune cells to support cancer growth. TEV miRNA cargo reprograms macrophage into tumour-associated macrophages (TAM). For example, miR-1246 and miR-940 (colon cancer-derived TEV), miR-222 (ovarian cancer), miR-29a-3p (oral squamous cell carcinoma), miR-21-5p (bladder cancer) and miR-503 (glioblastoma) were reported as essential players in TAM differentiation [197,198,199,200,201,202]. This results in the release of IL-10, TGFβ1 and matrix metalloproteinases by TAM to enhance tumour growth and progression. Likewise, MDSC, in response to TEV enriched in miR-34a, inhibit T-cell proliferation [203].

In addition to their canonical inhibitory function, miRNAs might also stimulate TLR7 and thus regulate target cell functions. For example, it was shown that TEV carrying miR-21 and miR-29a trigger NF-κB activation and the secretion of TNFα and IL-6 by binding to TLR7/8 [204]. A similar mechanism was described for miR-25 and miR-92a carried by liposarcoma-derived TEV [205]. Since the binding of TLR7 to the RNA is sequence specific [206,207], macrophage activation by TEV is fine-tuned. 

As previously mentioned, TEV regulate the maturation state of APC by suppressing the expression of co-activation and maturation markers. TEV-carried microRNAs and proteins play a part in this process. It was shown that TEV-miR-424 content induces resistance to immune checkpoint blockade by suppressing the CD28-CD80/86 costimulatory pathway in tumour-infiltrating DC and T cells [208]. miR-203 enriched in TEV from pancreatic tumours inhibits the expression of TLR4 in recipient DC [209], whereas miR-212 modulates the expression of MHC II [210]. Thus, under the control of TEV-miRNA content, APC acquire a suppressive phenotype and inhibit T-cell cytotoxic activity. TEV containing miRNA can also directly modulate T cells. Indeed, TEV-miR-214 content significantly downregulated PTEN expression in T cells and promoted their shift towards Treg in Lewis lung cancer [211]. 

TEV carry a pattern of long non-coding RNA (lncRNA) specific for each type of cancer. Circulating TEV from patients with pancreatic cancer contain several lncRNAs (i.e., FGA, KRT19, HIST1H2BK, ITIH2, MARCH2, CLDN1, MAL2, TIMP1, lnc-sox2ot) that correlate with the stage of the disease [212] and undergo downregulation after treatment [213]. The LncRNA PVT1 is commonly co-amplified with c-Myc. It encodes 52 ncRNAs variants, including 26 linear and 26 circular isoforms, and 6 microRNAs. PVT1 was associated with cancer immune regulation [214,215]. In addition, it was shown that in MDSC, PVT1 is upregulated by HIF-1α and modulates ARG1 activity and reactive oxygen species production, thereby suppressing T-cell antitumour immune response [216]. Overexpression of PVT1 in specific immune cells not only correlates with the suppression of antitumour response, but also contributes to treatment resistance, particularly immune checkpoint-based regimens [214]. In particular, high PVT1 expression predicts shorter disease-free survival in acute myeloid leukemia [217], ovarian [218], cervical [219] and colorectal cancer [220] and diffuse large B-cell lymphoma [221]; therefore, it has been proposed as a mechanism of chemo resistance. PVT1 was found in TEV of different cancers [222,223] and contributes to the exosome secretion of tumour cells [224]. 

As reported above, TEV stimulate immune cells to release their own TEV to suppress the immune response, thereby promoting tumour outgrowth. For example, TAM-derived TEV enhance the aerobic glycolysis and apoptotic resistance of breast cancer cells via myeloid-specific lncRNA, named HIF-1α-stabilising long non-coding RNA (HISLA) [225]. HISLA was found highly expressed in breast cancer patients and undergoes reduction in patients with remission, and therefore it has been proposed as biomarker of drug resistance [225]. 

## 4. TEV and Drug Resistance 

During treatment, tumour cells develop resistance through different mechanisms specifically impairing drug efficacy. In particular, drug efflux and inactivation [226], pro-survival pathway activation [227], EMT [228], immune evasion and de-differentiation towards stem-like phenotype [229] are the most relevant mechanisms. Drug efflux was detected in many types of cancer cells and depends on several transporters [230]. Among them, P-glycoprotein (p-gp) is the most investigated in cancer [231]. Physiologically, p-gp pumps out a broad range of xenobiotics and manages the efflux of toxins in the intestine, liver or kidney. Different types of cancer also exploit p-gp to decrease local drug concentration. The overexpression of p-gp was shown in haematological malignancies [232], colorectal cancer [233] and others [234]. Mechanisms of TEV-mediated acquired drug resistance mainly rely on protein or RNA transfer from resistant cells to susceptible recipient cells, or by exporting cytotoxic substances. As shown for doxorubicin-resistant osteosarcoma, the transfer of p-gp by TEV allows recipient cells to remove anticancer drugs [235]. Moreover, it was shown that TEV could transfer miR-451 and miR-27a to increase p-gp expression [236,237].

TEV cargo also turns on pro-survival pathways in recipient cancer cells. In particular, IL-6 promotes drug resistance [238] by upregulating multidrug resistance-related genes (MDR1 and GSTpi), apoptosis inhibitory proteins (Bcl-2, Bcl-xL and XIAP) and activating the Ras/MEK/ERK and PI3K/Akt signalling pathway [239]. Indeed, several studies demonstrated that TEV-mediated IL-6 production is a common mechanism of drug resistance [240,241,242]. The in vivo blockade of the release of TEV from TAM was reported as the most relevant mechanism improving gemcitabine sensitivity. The authors also provided evidence that TEV miR-365 content was involved in gemcitabine resistance [243]. 

As shown by Pavlyukov and colleagues [244], drug resistance also relies on RNA splicing. They found that glioblastoma cells release TEV enriched in the splicing factor RBM11. RBM11 is generally upregulated in apoptotic tumour cells during anticancer therapies. Once transferred to recipient cells, RBM11 switches the splicing of MDM4 and Cyclin D1 genes towards more oncogenic isoforms [244]. 

TEV also regulate immune checkpoint molecules to drive immune evasion, leading to tumour resistance. The expression of CD28 and CD80/86, required for PD-1 or CTLA-4 immune checkpoint blockade [245,246], was found under the control of TEV-miRNA content. Indeed, it was shown that miR-424 in TEV suppresses the expression of CD28-CD80/86 in TILs and DC, leading to drug resistance [208]. 

TEV surface proteins can act as decoy receptors for antibodies-based therapy, thereby decreasing drug bioavailability. In B-cell lymphoma, the expression of CD20 in TEV protects lymphoma cells from rituximab [247]. Similarly, in breast cancer, HER2+ TEV modulate resistance to the anti-HER2 monoclonal antibody [248]. In melanoma patients, circulating PD-L1+ TEV early after immune checkpoint blockade classifies patients as resistant to anti-PD-1 therapy [122]. Indeed, PD-L1+ TEV act as scavengers for immunotherapeutics [114,122]. 

Therefore, TEV-mediated transfer of proteins or RNAs induces drug resistance by reprogramming tumour cells towards a highly metastatic phenotype or by inactivating immune cells. 

## 5. TEV Released by TME Cells

Tumour tissue consists of cancer cells and recruited stromal, vascular and immune cells which, by releasing stimulatory growth factors, chemokines, and cytokines at the primary tumour site, promote angiogenesis and metastasis and suppress the anticancer immune response [249]. These cells in the TME include cancer-associated fibroblasts [240], tumour endothelial cells (TEC) [250], adipocytes [251,252], tumour-associated immune cells as TAM [253] and MDSC [254], DC [255], monocytes [256], CTL [257] and Treg [258], B cells [259], regulatory NK [260] and tumour-educated platelets [261]. Growing evidence indicates that cancer cells could recruit and reprogram all cell types. Notably, it was shown that TEV can induce tumour innervation [262] and that tumour-associated neuronal cells regulate tumour-infiltrated immune cell functions. 

TEV released from cancer-associated cells were described as tumour promoters. It has been shown that plasma from cancer patients contains a significantly higher EV concentration [263], originating from both tumour and normal cells. Herein, the immunosuppressive mechanisms involving circulating TEV derived from different tumour-associated cells are described (Figure 4). 

### 5.1. TEV Derived from Tumour-Educated Platelets

Activated platelets professionally release EV [264], which are the most abundant population of circulating EV. It has been shown that platelet concentration positively correlates with cancer growth [265]. Platelets and their EV are enriched in transcription factors, RNA and spliceosomes [266]. In cancer, megakaryocytes and platelets cross-talk with tumour via EV. This results in tumour-specific rearrangement of platelet transcriptome and proteome, leading to the establishment of the *so-called* tumour-educated platelets [267,268,269,270,271,272]. TEV expressing P-selectin (CD62P) and integrin αIIβ3 [273,274] regulate the immune system, stimulate angiogenesis, promote metastasis and are involved in drug resistance by activating discrete signalling cascades [275,276]. In particular, TEV released from tumour-educated platelets stimulate MAPK p42/44 and AKT, increase the expression of MMP1, MMP9, VEGF and cyclin D2 [277] and induce EMT by activating TGFβ1 and NF-κB signalling [278].

### 5.2. TEV from TEC

TEV derived from TEC are involved in cancer development, metastatic spread and tumour immune editing [39,279,280,281]. TEC-derived TEV contain pro-oncogenic proteins and RNAs involved in immune regulation (TGFβ1, HLA-G, IL-6, M-CSF, as well as lncRNA MALAT1 and miR-24-3p) [39,279,280]. It has been shown that TEC-derived TEV enhance the expression of PD-L1 in tumour and myeloid cells, promote the formation of Treg and decrease T-cell cytotoxic activity [39,279,281]. Interestingly, stress conditions in the TME (hypoxia, inflammation, changes in the pH and many others) induce the release of TEV with specific tumour-promoting properties. In particular, IL-3 signalling in TEC rearranges TEV miRNA cargo, which control β-catenin and PD-L1 expression as well as the metastatic spread [279,281].

### 5.3. EV Derived from DC

Tumour-primed immune cells can secrete immunologically active TEV. Commonly, EV derived from DC (DC-EV) carry functional MHC-complexes, thereby educating additional APC to recognise specific antigens [282] and cross-reactivating CD8+ T cells [283]. DC-EV play a critical role in generating antitumour CD8+ T cells. Indeed, several clinical trials based on tumour-primed DC-TEV were performed to find an effective antitumour vaccine [284,285]. 

However, growing evidence indicates that TEV mainly mediate tumour immune suppression by reprogramming cancer-infiltrated DC towards a tumour promoter phenotype. Thus, DC-mediated cross-presentation of tumour antigens commonly induces T-cell tolerance instead of antitumour activity.

DC-EV function depends on their molecular composition, in particular the concentration of co-stimulatory molecules (CD86, CD80, CD40) and antigen-presenting molecules (MHC-II, MHC-I) [286,287]. It was previously described that TEV cargo (HSP, TLR, HLA g [29]; S100A8, S100A9, Annexin A1 [30]; PGE2 [288]; TGFβ1 [289]; or microRNAs [290]), suppress DC maturation, resulting in the loss of tumour cell recognition [29,30]. Moreover, once activated by TEV, DC release their own TEV conveying tumour-associated antigens [291], which translate in T-cell cross-activation, antigen presentation, and immune tolerance [292,293]. Overall, TEV released by “immature” tumour-primed DC support cancer outgrowth by reprogramming immune cells towards immune tolerance. 

## 6. Conclusions

Early after EV discovery [294] it was reported that cancer cells release TEV enriched in proteins and lipids, contributing to tumour immune escape [295]. Soon after, the same group showed that TEV express antigens recapitulating their cell of origin and can exert both pro- or antitumour action [296]. Moreover, since MHC class II on TEV induces specific T-cell responses [297], TEV have been proposed for immunotherapeutic strategies. [4,282,291,298,299,300]. However, TEV-based anticancer vaccines still lack robust clinical benefits, since the complex TEV molecular composition mainly reprograms immune cells for tolerance [7,301]. After TEV-mediated activation, immune cells release their own TEV to amplify and spread the oncogenic signals. In fact, elevated circulating EV displaying oncogenic properties are commonly detected in cancer patients and contribute to drug resistance.

Overall, TEV may be considered a “complex system”, and prediction of their biological functions based on their individual molecular components may somehow be misleading. Therefore, efforts to develop methodologies to deeply investigate how their molecular components control the antitumour immune response in concert are warranted. If successful, EV engineering approaches would offer the chance to reshape the immune reaction into tumour recognition and eradication.

## Figures and Tables

**Figure 1 cancers-14-04020-f001:**
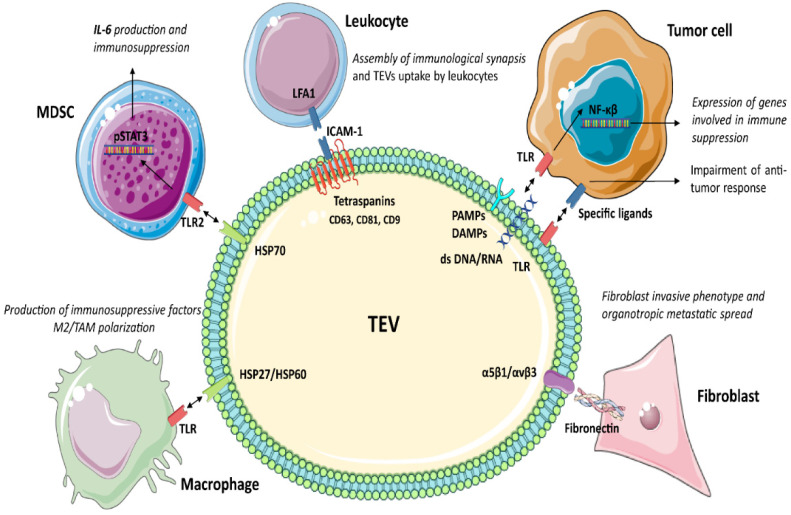
TEV markers relevant for immune regulation. TEV from cancer cells regulate the immune system by acting on different cell types. In particular, TEV express a large subset of molecules able to trigger tumour immune evasion. Indeed, the interaction between ICAM-1, embedded within tetraspanin domains (CD63, CD81 and CD9) in TEV and LFA1 in leucocytes, is essential for TEV uptake by leucocytes. Moreover, the cross talk between TEV DAMPs/PAMPs and dsDNA/RNA molecules and TLR in tumour cells stimulates the expression of genes involved in immune suppression. However, TLR can be also carried by TEV initiating signalling that impairs the antitumour immune response. Furthermore, HSP70 in TEV interacts with TLR2 on MDSC, boosting p-STAT3 expression and IL-6 production, while HSP27/60 interacting with TLR in TAM promotes differentiation of pro-tumoural M2 and the production of immune suppressive factors. α5β1 and αvβ3 integrins in TEV interact with fibronectin on fibroblasts and mediate signalling in recipient cells, supporting organotrophic metastatic spread. Licensed under a Creative Commons Attribution 3.0 Unported License. https://smart.servier.com (access on 15 August 2022).

**Figure 2 cancers-14-04020-f002:**
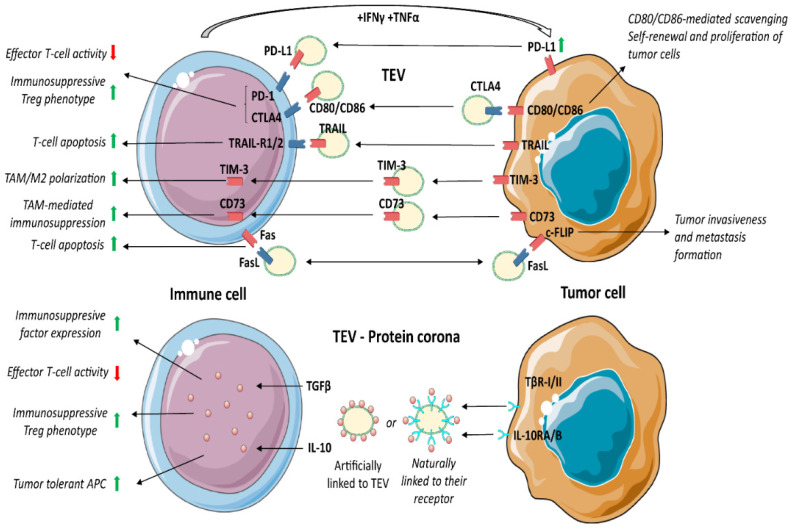
TEV as immunosuppressive shuttles. TEV from tumour cells carry various immune checkpoint molecules, which impair the antitumour immune response. In particular, activated T-cell producing IFNγ and TNFα enhance PD-L1 expression in tumour cells, which in turn secrete TEV expressing PD-L1, which interacting with PD-1, interferes with effector T-cell activity and drives Treg expansion. As PD-1, CTLA4 after binding to TEV expressing CD80/CD86 suppresses the immune response and T-cell-mediated cytotoxic activity. On the other hand, TEV carrying CTLA4 boost CD80/CD86 scavenging and induce proliferation of targeted tumour cells. Moreover, TIM-3+ TEV are uptaken by TAM and induce M2 polarisation. Similarly, CD73+ TEV upregulate the expression of immunosuppressive factors in TAM by activating the NF-κB pathway. Furthermore, FasL and TRAIL TEV content has immunosuppressive factors, by binding receptors Fas and TRAIL-R1/2, respectively, to immune cells, thereby promoting target cell apoptosis. In addition, FasL interaction with c-FLIP expressed by tumour cells promotes tumour invasiveness and metastasis formation. The so-called “Protein corona” describes the presence of TEV membrane proteins artificially linked to the membrane of TEV or naturally linked to their receptors. IL-10 and TGFβ expressed by TEV boost the immune suppression. https://smart.servier.com (access on 15 August 2022).

**Figure 3 cancers-14-04020-f003:**
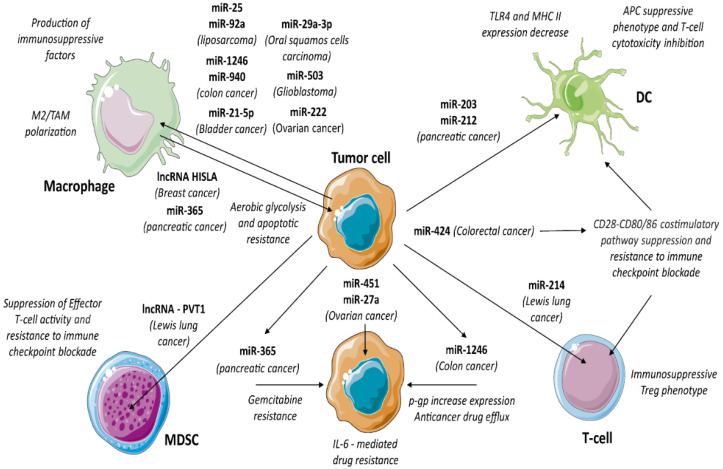
TEV non-coding RNA content. TEV contain and deliver a large subset of microRNAs and lncRNAs involved in immune suppression. In particular, specific miRNAs act on specific cell type. Indeed, miR-1246 and miR-940 (colon cancer), miR-222 (ovarian cancer), miR-29a-3p (oral squamous cell carcinoma), miR-21-5p (bladder cancer), miR-503 (glioblastoma), miR-25 and miR-92a (liposarcoma) stimulate macrophages to shift towards M2 phenotype. On the other hand, TEV derived from M2 macrophage carrying miR-365 (pancreatic cancer) and lncRNA HISLA (breast cancer) promote aerobic glycolysis and resistance to apoptosis in tumour cells. miR-424 (colorectal cancer) induces resistance to immune checkpoint blockade in tumour infiltrating DC and T-cells. Interestingly, miR-203 (pancreatic cancer) inhibits the expression of TLR4, while miR-212 modulates the expression of MHC II in recipient DC, which in turn induces a suppressive phenotype and inhibits T-cell cytotoxic activity. miR-214 (Lewis lung cancer) promotes the shift of T-cells towards Treg supporting immune suppression. Moreover, lncRNA-PVT1 (Lewis lung cancer) contributes to suppression CTL and resistance to immune checkpoint blockade. However, miRNAs also play a crucial role in drug resistance. Thus, miR-451 and miR-27a (ovarian cancer) enhance IL-6-mediated drug resistance, miR-365 (pancreatic cancer) induces gemcitabine resistance, while miR-1246 (colorectal cancer) upregulates p-gp to boost drug efflux. https://smart.servier.com (access on 15 August 2022).

**Figure 4 cancers-14-04020-f004:**
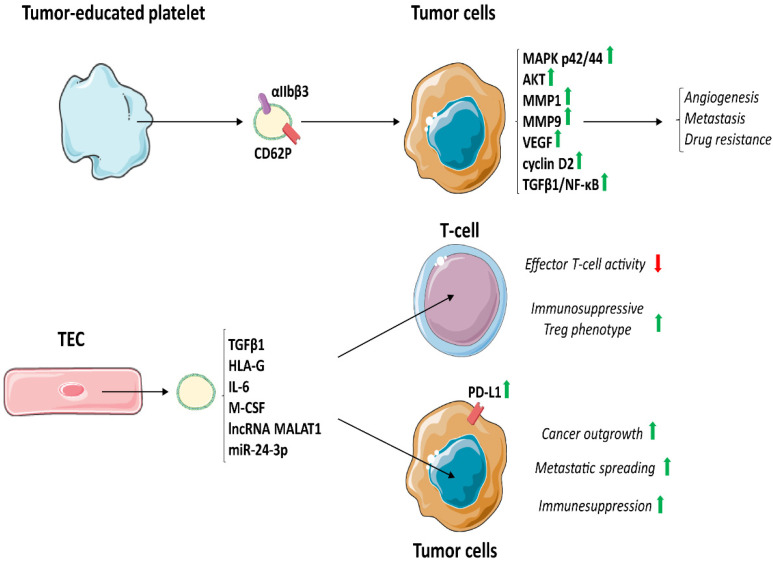
TEV released by tumour-educated platelets and tumour endothelial cells. Tumour-educated platelets produce TEV, expressing P-selectin (CD62P) and integrin αIIbβ3, stimulate MAPK p42/44 and AKT, and increase the expression of MMP1, MMP9, VEGF and cyclin D2. Moreover, TEV induce EMT by activating TGFβ1 and NF-κB signalling in tumour cells. All these signals increase angiogenesis, metastasis formation and drug resistance. On the other hand, TEC-derived TEV contain pro-oncogenic proteins and RNAs involved in immune regulation (TGFβ1, HLA-G, IL-6, M-CSF, as well as lncRNA MALAT1, and miR-24-3p), which, by increasing PD-L1 in tumour cells, boost cancer outgrowth, metastatic spread and immunosuppression. https://smart.servier.com (access on 15 August 2022).

## Data Availability

The data presented in this study are available in this article.

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
