# Peer review of "Tumour Derived Extracellular Vesicles: Challenging Target to Blunt Tumour Immune Evasion"

_cancers, 2022, doi:10.3390/cancers14164020_

Round 1

Reviewer 1 Report

This present review article by Lopatina et al described the involvement of tumor derived extracellular vesicles in immune evasion.  Also they address the importance of TEVs in tumor detection and their application in devising treatment against tumors. I am in principle supportive of accepting this work for publication. However, I have few suggestions to improve the manuscript for publication. 

Major 

  1. I strongly encouraged authors to use table format to layout ligand and receptors and their role in immune evasion with references.

Minor 

Ample of spelling mistakes was detected through the text,so, I encouraged authors to proof read the review carefully.

Author Response

We thank the Reviewer for His/Her positive comments.

I strongly encouraged authors to use table format to layout ligand and receptors and their role in immune evasion with references.

Response. As kindly suggested by the Reviewer a Table listing ligands, receptors and their role in immune evasion has been added. The Table also includes referred References.

Ample of spelling mistakes was detected through the text, so, I encouraged authors to proof read the review carefully.

Response. As kindly suggested by the Reviewer, the text has been edited

Reviewer 2 Report

The review “Tumour derived extracellular vesicles: challenging target to blunt tumour immune evasion” by Lopatina et al is focused on the role of tumor extracellular vesicles in cross-talk between tumor cells and the cells of microenvironment in the light of their influence on predominantly protumor immunity. The task of the authors was not an easy one; during last decade aprox. 3000 reviews describing the tumor derived EVs were published, many of them were about tumor-TME cross-talk, including involvement of pro- and anti-tumor immunity. Did the authors of this review manage to say something new? Especially considering the fact that the number of experimental articles in this area slightly outnumbered by the number of reviews. 

In this review, the authors thoroughly consider the role of each protein constituting the EVs, including tetraspanins, Toll-like receptors, heat shock proteins, program death receptor 1, CTLA-4, TIM-3 and so on. 

In general, the review is clearly written and nicely illustrated, however, the impression remains of some superficiality and careless in the citation of the literature. I wish to focus my criticism on chapter 3.3 "Heat shock proteins". 

Firstly, it is not always possible to understand which particular HSP, Hsp27, Hsp70 or Hsp90 refers to this or that point. The chaperones belonging to above families have quite heterologous structures and different functions inside cells and in the extracellular space. Secondly, all they can be released from cells both in the form of EVs and in a soluble form, and authors often confuse the effects of EVs enriched with HSP and of free HSP. For example, the statement that “a long-lasting HSP-mediated anti-tumour response may translate in immune tolerance” does not refer to TEVs. The article [91] refers to soluble Hsp70, as well as papers in refs [93, 94] reveal the effects of soluble Hsp27 and Hsp90. Telling about that “HSP+ TEV can act as immunostimulatory and immunosuppressive controllers” authors refer to articles by Ullrich et al., 1986 [89] (probably in benefit of TEVs’ anti-tumor activity and Chalmin et al., 2010 [92] favoring TEVs’ protumor activity. Indeed, the paper by Chalmin et al, presents TEVs with membrane Hsp70 (not HSP+TEVs!) and the paper by Ullrich et al is about a mouse tumor-specific transplantation antigen that was explored as heat shock-related protein, and there was no word about TEVs and their anti-tumor immunity. Meanwhile, there is a significant body of data about Hsp70-bearing TEVs demonstrating their controversial role in tumor progression. First, tumor-derived exosomes with membrane-bound Hsp70 were shown to increase the immunosuppressive activity of myeloid-derived suppressor cells (MDSC) via activation of STAT3 (Chalmin, F. et al. 2010, J. Clin. Invest; Diao, J. et al. 2015; Med. Oncol).

Secondly, exosomes secreted under hypoxia with a high content of Hsp70, enhanced the invasiveness and stemness of prostate cancer cells (Ramteke, A. et al., 2015, Mol. Carcinog).

On the contrary, the antitumor activity of Hsp70-bearing EVs from heat-treated CT-26 and B16 cells was demonstrated, which produced a strong Th1 immune response and tumor eradication in allogeneic hosts in vivo (Cho, J. A., et al., Cancer Lett). Additionally, the strong anti-tumor effects of exosomes from heated mouse colon adenocarcinoma MC38 cells were accompanied by the conversion of regulatory T cells into Th17 cells in an IL6-dependent manner (Guo, D. et al., 2017, Immunology).

Hsp70-positive exosomes from resistant to chemotherapy HepG2 cells showed immunoregulatory properties by stimulating an NK-mediated antitumor response and by the enhanced production of granzyme B (Lv, L.-H. et al., 2012, J. Biol. Chem). Finally, TEVs from B16 mouse melanoma and CT-26 mouse colon carcinoma, enriched with Hsp70 suppressed tumor growth in vivo, significantly increasing animals surviving, by elevating the CD8+ lymphocytes number in a line with their cytotoxic anti-tumor power and diminishing TAMs’ number in tumor lesion (Komarova et al., 2021, Sci.Rep.). Thus, besides the report of Ullrich et al., there is a number of studies that could be discussed by the authors.   

Moreover, I found almost no data concerning immunostimulatory role of Hsp90-, Hsp27-, and Hsp105-containg TEVs while the report of [95] concerned Hsp90 in circulating form. 

This overview demonstrates the luck of knowledge in a particular field of cancer EVs containing HSP which does not allow me to recommend the m/s of Lopatina et al for publication in Cancers in present form

Minor: 

1.There are flaws in the manuscript design, important sections are missed, such as Funding, Acknowledgement, Conflict of interests, Author contribution.

2. Ref [90] is in the Reference list but is not cited in the text

Author Response

We thank the Reviewer for His/Her constructive comments.

The review “Tumour derived extracellular vesicles: challenging target to blunt tumour immune evasion” by Lopatina et al is focused on the role of tumor extracellular vesicles in cross-talk between tumor cells and the cells of microenvironment in the light of their influence on predominantly protumor immunity. The task of the authors was not an easy one; during last decade aprox. 3000 reviews describing the tumor derived EVs were published, many of them were about tumor-TME cross-talk, including involvement of pro- and anti-tumor immunity. Did the authors of this review manage to say something new? Especially considering the fact that the number of experimental articles in this area slightly outnumbered by the number of reviews. 

In this review, the authors thoroughly consider the role of each protein constituting the EVs, including tetraspanins, Toll-like receptors, heat shock proteins, program death receptor 1, CTLA-4, TIM-3 and so on. 

In general, the review is clearly written and nicely illustrated, however, the impression remains of some superficiality and careless in the citation of the literature. I wish to focus my criticism on chapter 3.3 "Heat shock proteins". 

Firstly, it is not always possible to understand which particular HSP, Hsp27, Hsp70 or Hsp90 refers to this or that point. The chaperones belonging to above families have quite heterologous structures and different functions inside cells and in the extracellular space. Secondly, all they can be released from cells both in the form of EVs and in a soluble form, and authors often confuse the effects of EVs enriched with HSP and of free HSP. For example, the statement that “a long-lasting HSP-mediated anti-tumour response may translate in immune tolerance” does not refer to TEVs. The article [91] refers to soluble Hsp70, as well as papers in refs [93, 94] reveal the effects of soluble Hsp27 and Hsp90. Telling about that “HSP+ TEV can act as immunostimulatory and immunosuppressive controllers” authors refer to articles by Ullrich et al., 1986 [89] (probably in benefit of TEVs’ anti-tumor activity and Chalmin et al., 2010 [92] favoring TEVs’ protumor activity. Indeed, the paper by Chalmin et al, presents TEVs with membrane Hsp70 (not HSP+TEVs!) and the paper by Ullrich et al is about a mouse tumor-specific transplantation antigen that was explored as heat shock-related protein, and there was no word about TEVs and their anti-tumor immunity. Meanwhile, there is a significant body of data about Hsp70-bearing TEVs demonstrating their controversial role in tumor progression. First, tumor-derived exosomes with membrane-bound Hsp70 were shown to increase the immunosuppressive activity of myeloid-derived suppressor cells (MDSC) via activation of STAT3 (Chalmin, F. et al. 2010, J. Clin. Invest; Diao, J. et al. 2015; Med. Oncol).

Secondly, exosomes secreted under hypoxia with a high content of Hsp70, enhanced the invasiveness and stemness of prostate cancer cells (Ramteke, A. et al., 2015, Mol. Carcinog).

On the contrary, the antitumor activity of Hsp70-bearing EVs from heat-treated CT-26 and B16 cells was demonstrated, which produced a strong Th1 immune response and tumor eradication in allogeneic hosts in vivo (Cho, J. A., et al., Cancer Lett). Additionally, the strong anti-tumor effects of exosomes from heated mouse colon adenocarcinoma MC38 cells were accompanied by the conversion of regulatory T cells into Th17 cells in an IL6-dependent manner (Guo, D. et al., 2017, Immunology).

Hsp70-positive exosomes from resistant to chemotherapy HepG2 cells showed immunoregulatory properties by stimulating an NK-mediated antitumor response and by the enhanced production of granzyme B (Lv, L.-H. et al., 2012, J. Biol. Chem). Finally, TEVs from B16 mouse melanoma and CT-26 mouse colon carcinoma, enriched with Hsp70 suppressed tumor growth in vivo, significantly increasing animals surviving, by elevating the CD8+ lymphocytes number in a line with their cytotoxic anti-tumor power and diminishing TAMs’ number in tumor lesion (Komarova et al., 2021, Sci.Rep.). Thus, besides the report of Ullrich et al., there is a number of studies that could be discussed by the authors.   

Moreover, I found almost no data concerning immunostimulatory role of Hsp90-, Hsp27-, and Hsp105-containg TEVs while the report of [95] concerned Hsp90 in circulating form. 

This overview demonstrates the luck of knowledge in a particular field of cancer EVs containing HSP which does not allow me to recommend the m/s of Lopatina et al for publication in Cancers in present form

Response:  We thank the Reviewer for appropriate comments, since they have undoubtedly improved the Ms. The Reviewer’s suggestions have been included and the specific paragraph revised (the main revised paragraphs and sentences have been highlighted in green).

Minor: 

1.There are flaws in the manuscript design, important sections are missed, such as Funding, Acknowledgement, Conflict of interests, Author contribution.

Response. We apology, these sections have been filled

  1. Ref [90] is in the Reference list but is not cited in the text

Response. We apology, all references have been updated

Round 2

Reviewer 2 Report

The authors took into account the comments of the reviewer and added the necessary information to the text. I think that in its present form the article can be accepted for publication in Cancers